# Iatrogenic Ankle Charcot Neuropathic Arthropathy after Spinal Surgery: A Case Report and Literature Review

**DOI:** 10.3390/medicina58121776

**Published:** 2022-12-02

**Authors:** Sung Hwan Kim, Woo-Jong Kim, Eun Seok Park, Jun Yong Kim, Young Koo Lee

**Affiliations:** 1Department of Orthopaedic Surgery, Soonchunhyang University Hospital Bucheon, 170, Jomaru-ro, Wonmi-gu, Bucheon-si 14584, Republic of Korea; 2Department of Orthopaedic Surgery, Soonchunhyang University Hospital Cheonan, 31, Sooncheonhyang 6-gil, Dongnam-gu, Cheonan 31151, Republic of Korea

**Keywords:** ankle, charcot neuropathic arthropathy, iatrogenic charcot, spinal surgery

## Abstract

Charcot neuropathic arthropathy is a relatively rare, chronic disease that leads to joint destruction and reduced quality of life of patients. Early diagnosis of Charcot arthropathy is essential for a good outcome. However, the diagnosis is often based on the clinical course and longitudinal follow-up of patients is required. Charcot arthropathy is suspected in patients with suggestive symptoms and an underlying etiology. Failed spinal surgery is not a known cause of Charcot arthropathy. Herein we report a patient with ankle Charcot neuropathic arthropathy that developed after failed spinal surgery. A 58-year-old man presented to the emergency room due to painful swelling of the left ankle for 2 weeks that developed spontaneously. He underwent spinal surgery 8 years ago that was associated with nerve damage, which led to weakness of great toe extension and ankle dorsiflexion, and sensory loss below the knee. CT and T2-weighted sagittal MRI showed a fine erosive lesion, subluxation, sclerosis, fragmentation, and large bone defects. Based on the patient’s history and radiological findings, Charcot arthropathy was diagnosed. However, the abnormal blood parameters, positive blood cultures, and severe pain despite the decreased sensation suggested a diagnosis of septic arthritis. Therefore, diagnostic arthroscopy was performed. The ankle joint exhibited continued destruction after the initial surgery. Consequently, several repeat surgeries were performed over the next 2 years. Despite the early diagnosis and treatment of Charcot arthropathy, the destruction of the ankle joint continued. Given the chronic disease course and poor prognosis of Charcot arthropathy, it is essential to consider this diagnosis in patients with neuropathy.

## 1. Introduction

Charcot neuropathic arthropathy was first reported by Jean-Martin Charcot in 1868 as a progressive joint disease characterized by gradual joint destruction [1]. It leads to painful or painless destruction of bones and joints in patients with neuropathy [2]. The pathophysiology of Charcot neuropathic arthropathy involves sensorineural, autonomic, and motor dysfunction, which lead to joint instability, osteopenia, microtrauma [3], acute localized inflammation, and bone destruction (e.g., subluxation, dislocation, and deformity) [4]. The joint deformity prevents the use of standard footwear and results in ulceration, deep infections, and even amputation [5]. The leading cause of Charcot arthropathy in previous decades was neurosyphilis; currently, the leading cause is diabetes mellitus [6]. Charcot arthropathy is often a chronic complication of diabetes mellitus with or without polyneuropathy [7]. Additionally, Charcot arthropathy may be caused by leprosy, spinal anesthesia, and spinal diseases [8]. A recent case report described a patient who developed knee Charcot arthropathy after spinal canal surgery [9]. However, there are no previous reports of ankle Charcot arthropathy caused by spinal surgery. Herein we describe a patient who developed ankle Charcot arthropathy after iatrogenic trauma during spinal surgery.

## 2. Case Presentation

### 2.1. Preoperative Evaluation

A 58-year-old man presented to the emergency room with painful swelling of the left ankle for 2 weeks that developed spontaneously. He had a history of spinal surgery and associated nerve damage 8 years ago, with residual weakness of great toe extension and ankle dorsiflexion and reduced sensation in the leg. He was able to ambulate for 50 m using a walker. The left ankle did not have a wound but was swollen, red, and warm (Figure 1). He complained of mild to moderate pain around the swollen left ankle. He had grade II great toe extension, ankle dorsiflexion and plantar flexion strength. Passive extension of the ankle was intact, with no limitation in the range of motion. There was partial sensory loss over the entire lower leg and foot. The patient had a sensory stimulus score of 2 below the ankle and 5 between the knee and ankle, compared to 10 at the normal. These deficits developed immediately after spinal surgery; the patient underwent rehabilitation therapy, which was unsuccessful. The blood glucose and HbA1c levels were 98 mg/dL (normal range: 60–99 mg/dL) and 5.5% (normal range: 4–6%), respectively. The erythrocyte sedimentation rate was 120 mm/hr (normal range: <5 mm/hr) and the C-reactive protein (CRP) level was 38.87 mg/dL (normal range: <0.5 mg/dL). A blood culture obtained in the emergency room showed the growth of methicillin-sensitive *Staphylococcus aureus* (MSSA). Radiographs showed erosion and subluxation of the distal tibiofibular joint and talus of both ankles (Figure 2). Computer tomography (CT) showed increased bone density around the ankle, indicating a chronic gliding mechanism. The tibial bone defect was similar in shape to the talar dome. In addition, multiple bony fragments were scattered in the distal tibia (Figure 3). T2-weighted coronal magnetic resonance imaging (MRI) revealed a cystic mass and joint destruction in the distal tibia with no erosion. There were no periarticular edema or bone marrow abnormalities (Figure 4). The patient was diagnosed with Charcot arthropathy based on the characteristic imaging findings of subluxation, sclerosis, fragmentation, and large bone defects. Although fine erosive lesions are less common in septic arthritis [2], the diagnosis of septic arthritis could not be excluded because of severe pain despite decreased sensations and abnormal blood parameters. Therefore, we performed arthroscopic surgery to exclude joint infection.

### 2.2. Surgical Procedure

Arthroscopic surgery was performed for examination and irrigation. Intraoperatively, bony fragments were scattered inside the ankle joint and the distal tibial articular surface was unevenly fragmented at the medial talar dome (Figure 5). The inflammatory tissue was debrided using a shaver and the free fragments were removed using forceps. The tissue culture obtained during surgery showed the growth of MSSA.

The CRP level remained high after the first surgery; therefore, the infectious diseases department was consulted and intravenous antibiotics were administered. Due to the persistently raised CRP level at 1 month after the first surgery, we performed a second surgery to insert an anti-bead and apply an external fixator. During the second surgery, soft tissue dissection revealed large quantities of fragile, chronic inflammatory tissue around the ankle (Figure 6). The tissue was removed by debridement and curettage. Then, an anti-bead was inserted and the Ilizarov apparatus was applied.

After the second operation, the patient was continued on intravenous antibiotics and the CRP level declined. Therefore, we performed anti-cement removal and tibiotalar fusion at 4 months after the second operation. After removal of the anti-bead inserted during the previous operation, rigid fixation using a locking screw and plate was performed, as well as a bone graft.

At the 1-year follow-up after the third operation, the patient exhibited talar subluxation at the site of ankle fusion. After discussion with the patient, subtalar fusion was performed. A bone graft was performed by harvesting auto bone in the left iliac area. The previously fused ankle joint showed talar subluxation. We performed decoration followed by grafting of the harvested auto-iliac bone. Staple and cannulated screws were used for rigid fixation. Finally, external fixation was applied using the Ilizarov fixator.

### 2.3. Postoperative Care

After the final surgery, the patient was regularly followed-up in the outpatient clinic; this is still ongoing. He has been advised to use a cam-walker and avoid weight bearing. The patient does not have significant pain but walking difficulty has persisted. The 1-year follow-up X-ray (Figure 7) showed no disruption of alignment but worsened bone collapse compared to the 2-month follow-up X-ray (Figure 8). The progressive bone collapse may require additional surgery. Although the patient has maintained alignment due to the avoidance of weight bearing, he may require amputation if subluxation or dislocation recurs.

## 3. Discussion

Charcot neuropathic arthropathy has several causes, of which the most common is diabetes mellitus [10]. Other causes include several unrelated diseases that are complicated by nerve injury, including infection-related distal neuropathies (e.g., leprosy and syphilis), diseases of the spinal cord and nerve roots (e.g., tabes dorsalis, trauma, and syringomyelia), systemic diseases (e.g., Parkinson’s disease, human immunodeficiency virus, sarcoidosis, rheumatoid disease, and psoriasis) [9], and toxins (e.g., ethanol and drug use) [11,12]. In our patient, the aforementioned causes were excluded based on the history, laboratory, and radiological findings. Our patient is similar to the previously reported case of knee Charcot neuropathic arthropathy that developed after nerve damage sustained during previous spinal surgery [7]. Our patient developed a superimposed infection that led to a high CRP level and growth of MSSA on the blood and tissue culture. The pathophysiology of Charcot neuropathic arthropathy involves increased blood flow to the bones due to damage to the sympathetic nerves, which results in bone resorption and weakening, ultimately leading to fractures and deformities [13]. Charcot neuropathic arthropathy is a chronic and progressive disease that is often difficult to diagnose [14]. The characteristic radiological findings of Charcot arthropathy include progressive bony destruction; however, there are no isolated laboratory or radiological findings that can confirm the diagnosis. Therefore, follow-up evaluation is often required [15]. Additionally, infection cannot be reliably excluded in cases with radiological findings of bony destruction. Therefore, laboratory and radiology examinations are often performed for patients with bony destruction [16]. In cases with infection, arthroscopy or incision and drainage and intravenous antibiotics, may be required. If the follow-up imaging reveals continued bone collapse despite no evidence of major trauma even after the infection has been treated, the possibility of Charcot neuropathic arthropathy should be considered [17]. The risk of Charcot arthropathy is particularly high in cases of neurological deficits, such as in our patient.

The limitation of this case report is that it describes a single case of Charcot arthropathy. Additionally, the pathophysiology of Charcot arthropathy was not explored. Despite early diagnosis and treatment of Charcot arthropathy, the disease continued to progress in our patient. The possibility of Charcot neuropathic arthropathy should be considered in patients with a history of neural trauma sustained during spinal surgery.

## 4. Conclusions

Spinal cord injury caused by neural trauma, such as failed spinal surgery, can cause Charcot neuropathic arthropathy. Therefore, such patients should be carefully evaluated for Charcot arthropathy, particularly in cases with severe bone collapse without a history of major trauma.

## Figures and Tables

**Figure 1 medicina-58-01776-f001:**
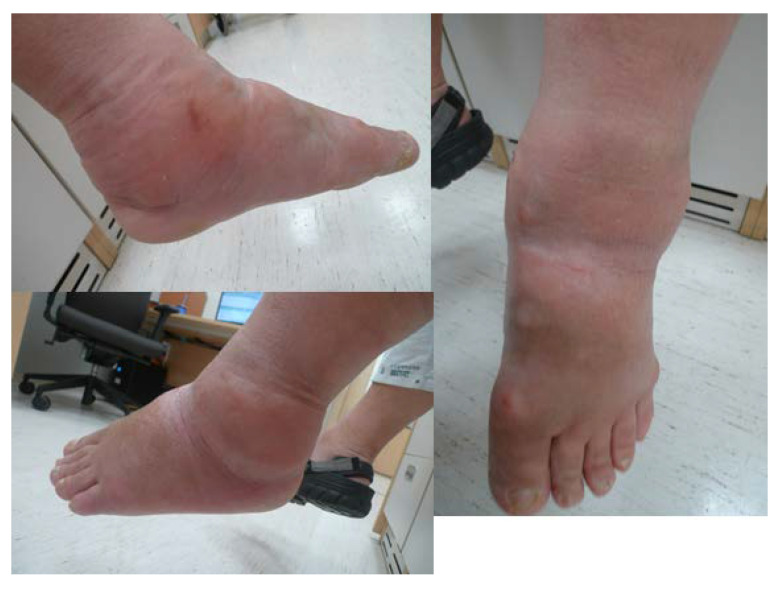
Left ankle at the time of presentation. There was no abrasion or laceration. The skin overlying the ankle joint was swollen and red.

**Figure 2 medicina-58-01776-f002:**
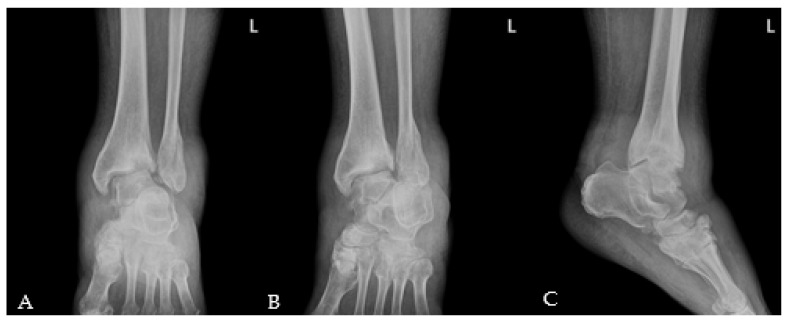
Initial left ankle X-ray. (**A**) Anteroposterior, (**B**) mortise, (**C**) lateral. Subluxations of the ankle joint and bony fragments were seen. An old healed fracture of the lateral malleolus was also visible.

**Figure 3 medicina-58-01776-f003:**
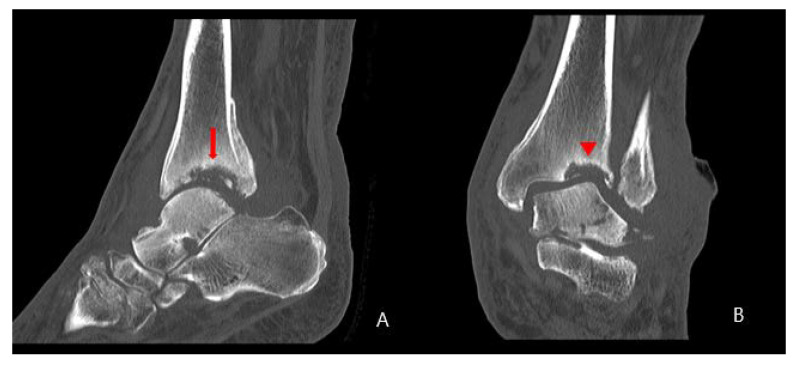
Initial left ankle: (**A**) sagittal and (**B**) coronal computer tomography scans. (**A**) Multiple bony fragments were scattered in the distal tibia (arrow). (**B**) The bone density was increased around the ankle and the tibial bone defect had a similar shape to the talar dome (arrowhead).

**Figure 4 medicina-58-01776-f004:**
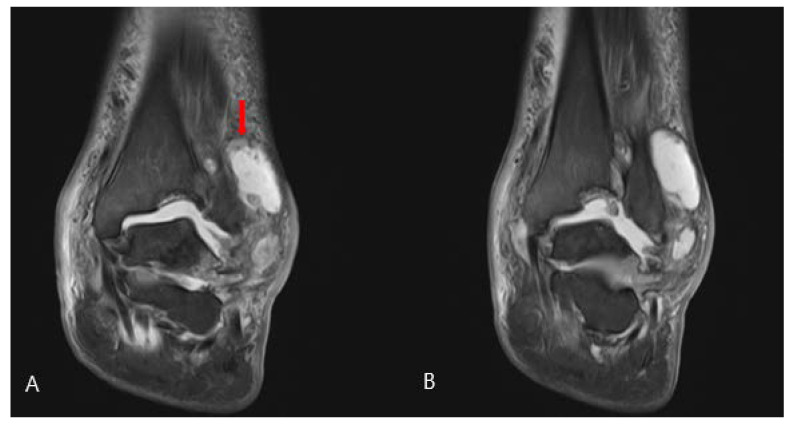
Initial left ankle T2-weighted coronal magnetic resonance imaging. (**A**) A cystic mass was seen near the distal fibula (arrow). (**B**) Joint destruction without bony erosion was seen in the distal tibia. There was no periarticular edema and the bone marrow was normal.

**Figure 5 medicina-58-01776-f005:**
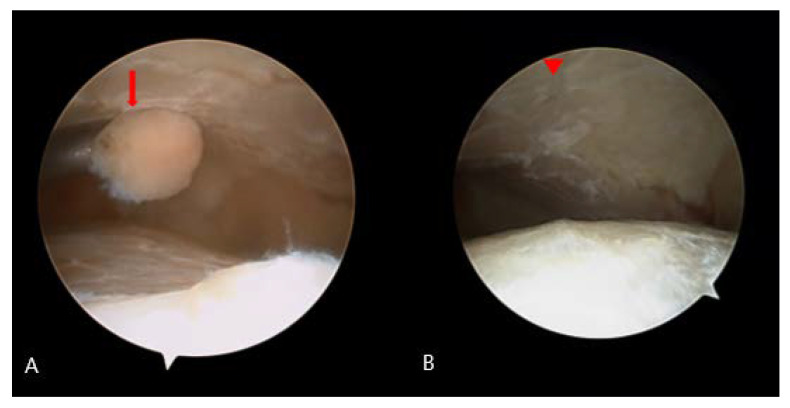
Findings during first arthroscopy. (**A**) Bony fragments were scattered inside the ankle. joint (arrow). (**B**) The distal tibial articular surface was unevenly fragmented at the level of the medial talar dome (arrowhead).

**Figure 6 medicina-58-01776-f006:**
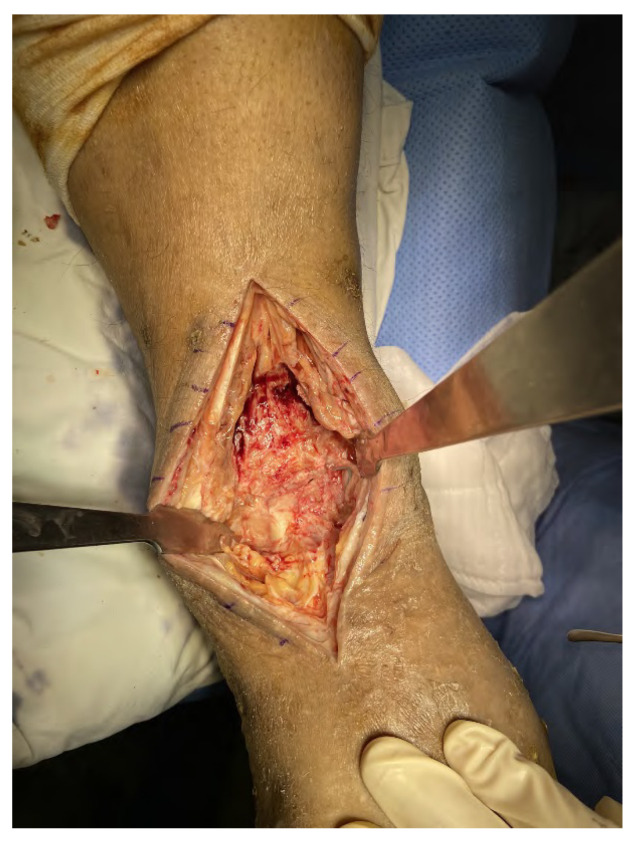
Intraoperative findings during the second surgery. A large quantity of fragile, chronic inflammatory tissue was observed around the ankle joint.

**Figure 7 medicina-58-01776-f007:**
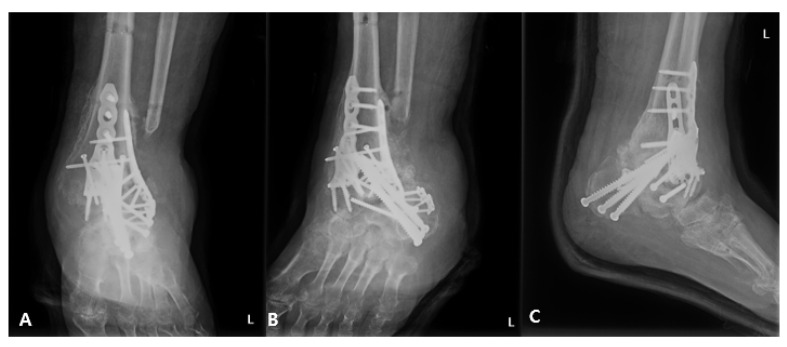
Left ankle X-ray. (**A**) Anteroposterior, (**B**) mortise, and (**C**) lateral images acquired 2 months after the final surgery. The images were obtained after removal of the external fixator.

**Figure 8 medicina-58-01776-f008:**
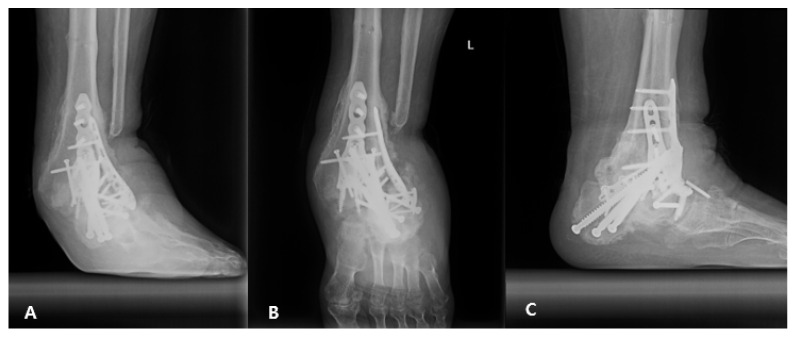
Left ankle X-ray. (**A**) Anteroposterior, (**B**) mortise, and (**C**) lateral images acquired 1 year after the final surgery. The X-ray showed progressive bone collapse but no worsening of alignment.

## Data Availability

Data sharing is not applicable to this article because any datasets were made or analyzed during this study.

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
