# Peer review of "Iatrogenic Ankle Charcot Neuropathic Arthropathy after Spinal Surgery: A Case Report and Literature Review"

_medicina, 2022, doi:10.3390/medicina58121776_

Round 1

Reviewer 1 Report

The manuscript is interesting in regard to the topic addressed. The metodological approach is correct and the conclusions are in line with the results obtained.

It could be worthy of publication, being very relevant to the important issue. It would be useful to mention PMID: 32913602  because it is one of the most recent literature review  

  •  
  •  

Reviewer 2 Report

It is an interesting and relevant topic. This paper is a good contribution about Charcot neuropathic arthropathy although the differentiation from the septic process is not definitely clear. The methods used and the conclusions are appropriate. The results and the discussion are presented clearly. I have no improvement suggestions. This is a good and interesting case report. The manuscript should be accepted.

Reviewer 3 Report

Title: Iatrogenic Ankle Charcot Neuropathic Arthropathy after Spinal Surgery: A Case Report and Literature Review

Comments: The case study report was explored in a very systematic order. All the procedures and management plan were elaborated in the case presentation including the follow-up images and management. The case report gives a good picture of the linking between the iatrogenic causes to the Charcot neuropathic arthropathy that was caused after the spinal surgery.

1.    Introduction – Adequate introduction to the Charcot Neuropathic arthropathy was given with a recent reference of a case report of a patient who developed Charcot arthropathy of the knee after spinal canal surgery. Though, ‘literature review’ should not be included in the title because the literature review in the paper is the minimal, enough for the introduction of the topic.

2.    Case presentation – The case presented provided adequate past medical history with evaluation and imaging reports. The surgical procedures were described with management that was done for the patient. Post-operative care was also describe with appropriate prognosis evaluated as well.

3.    Discussion – The discussion of the case report was good. The patient was discussed and compared with the case report on the Charcot arthropathy of the knee. The investigations and radiological imaging for the patient was described with recommendations if a similar case is presented.

4.    Limitations and Conclusion – The limitations of the study were described with a good conclusion which summarizes the entire case report
